# Progress and gaps in childhood immunization among two-year-olds in Ghana (1993–2022): A trend and equity analysis

Florence Gyembuzie Wongnaah[1]*, Augustus Osborne[2], Khadijat Adeleye[3], Bright Opoku Ahinkorah[4,5]

1 Abuakwa North Municipal Health Directorate, Ghana Health Service, Kukurantumi, Eastern Region, Ghana 2 Institute for Development, Freetown, Sierra Leone, 3 Elaine Marieb College of Nursing at the University of Massachusetts, Amherst, United States of America, 4 REMS Consultancy Services Limited, Sekondi-Takoradi, Western Region, Ghana, 5 College of Health, Adelaide University, Adelaide, Australia

* florencewongnaah@yahoo.com

## Abstract

### Background

Immunization remains a cornerstone of child survival and population health. While Ghana has made significant strides in vaccine delivery over the past three decades, gaps in equitable and universal coverage persist. This study examined the progress and gaps in full immunization coverage among two-year-olds in Ghana from 1993 to 2022.

### Methods

We used data from the Ghana Demographic Health Survey rounds conducted between 1993 and 2022 to examine full immunization coverage among two-year-olds. Disaggregated data were accessed via the WHO Health Equity Assessment Toolkit (HEAT). Inequality was assessed across six dimensions: maternal age, household wealth status, maternal education, place of residence, child's sex, and subnational region. Various inequality measures, including difference, ratio, population-attributable risk, and population-attributable fraction were calculated.

### Results

Full immunization coverage improved from 55.5% in 1993 to 71.5% in 2022, peaking at 77.6% in 2008. Inequality analysis showed reduced socioeconomic disparities related to wealth, maternal education, and urban-rural residence by 2022. However, two key inequalities persisted: maternal age-related inequality significantly widened to 26.8 percentage-point gap in 2022 between mothers aged 20–49 and adolescent mothers (15–19 years), and substantial regional disparities remained, with a significant 39.8 percentage-point gap between the best- and worst-performing regions.

**Data availability statement:** The data used in this study are publicly accessible through the World Health Organization Health Equity Assessment Toolkit (HEAT) platform, available at https://whoequity.shinyapps.io/heat/. Researchers can obtain the relevant datasets by visiting the platform, where detailed descriptions and titles of available data are provided. Access is available upon following the instructions outlined on the website. The datasets hosted on this platform are sufficient to replicate the analytical methods applied in this study and to verify the reported findings. The authors confirm that they did not receive any privileged or exclusive access to the data, and all datasets are openly available to researchers.

**Funding:** The author(s) received no specific funding for this work.

**Competing interests:** The authors have declared that no competing interests exist.

**Abbreviations:** D, Difference; GDHS, Ghana Demographic and Health Survey; HEAT, Health Equity Assessment Toolkit; DHS, Demographic Health Survey; PAF, Population Attributable Fraction; PAR, Population Attributable Risk; R, Ratio; WHO, World Health Organization

## Conclusion

Despite the gains in full immunization coverage in Ghana over the past three decades, significant inequalities persist, particularly among adolescent mothers and populations in disadvantaged regions. Strengthening equity-focused immunization strategies is essential to achieving universal coverage.

## Introduction

Immunization is a fundamental pillar of public health intervention, serving as a critical defence mechanism against preventable infectious diseases, particularly for vulnerable young children [1,2]. This significance is magnified by global demographics, with UNICEF reporting approximately 680 million children under five years worldwide, including 4 million in Ghana [3]. The period from birth to two years represents a crucial developmental window [4], during which children are especially susceptible to infectious diseases due to their developing immune systems [5,6]. This vulnerability, coupled with the proven effectiveness of early vaccination, underscores the urgent need for comprehensive immunization coverage during these formative years.

The World Health Organization and national health authorities have established robust vaccination protocols as one of the most cost-effective strategies for reducing childhood morbidity and mortality [7–9]. Ghana's implementation of the Expanded Program on Immunization (EPI) in 1978 marked a significant milestone in the nation's public health history, facilitating systematic vaccine delivery and substantially reducing preventable childhood diseases [10]. This program's implementation has been particularly crucial in addressing developing nations' complex health challenges, where infectious diseases pose significant threats to child survival and development.

The emphasis on immunization for children under two years is grounded in multiple scientific and public health rationales. Critical among these is the establishment of early immunity, which protects individual children and contributes to community-wide disease resistance through herd immunity [11]. Furthermore, early childhood vaccination aligns strategically with global health initiatives, particularly the Sustainable Development Goals (SDGs), specifically SDG 3, which focuses on ensuring healthy lives and promoting well-being for all ages [12,13]. Immunization directly contributes to several targets of SDG 3, including reducing child mortality, preventing the spread of infectious diseases, and ensuring universal health coverage [12]. By preventing diseases such as measles, polio, and hepatitis, immunization not only safeguards the health of children but also fosters healthier communities, thereby playing an integral role in achieving broader global health objectives and sustainable development targets [13]. While Ghana's EPI program has achieved remarkable progress, with national immunization coverage rates reaching 90–95% for most vaccines over the past decade, significant challenges persist [14]. These challenges are particularly evident in the disparities between urban and rural areas and among different socio-economic groups. The complexity of achieving and maintaining high immunization coverage is further complicated by geographical accessibility issues, healthcare

resource distribution inequities, and varying levels of health literacy among different population segments [15,16]. These factors contribute to creating pockets of under-immunized children, particularly in rural areas and urban slums, where healthcare access remains limited.

Recent epidemiological data reveals disparities in vaccination coverage across Ghana's regions, particularly for newer vaccines such as pneumococcal conjugate (PCV) and rotavirus [14,17]. These inequalities reflect systemic healthcare access issues and highlight the need for targeted interventions to reach underserved populations. This study specifically addresses the lack of long-term equity analyses in immunization coverage in Ghana, aiming to identify persistent gaps and disparities over time. Understanding the longitudinal trends in immunization coverage from 1993 to 2022 is crucial for several reasons: it provides insights into the effectiveness of past health interventions, identifies persistent coverage gaps, and informs evidence-based policy decisions. The objective of this study was to examine the progress and gaps in full immunization coverage among two-year-olds in Ghana from 1993 to 2022, with the aim of generating evidence to inform equitable strategies toward achieving and sustaining universal immunization coverage.

## Methods

### Study design and data source

This research used cross-sectional data from the Ghana Demographic and Health Surveys (GDHS) conducted between 1993 and 2022. These surveys are nationally representative household evaluations conducted periodically in Ghana. They collect data on many health indicators, including child growth and development. The GDHS utilises a multi-stage cluster sampling methodology. This involves dividing the nation into geographic areas (primary sampling units or PSUs), randomly selecting a subset of PSUs, and then identifying smaller geographic units (secondary sampling units or SSUs) inside each selected PSU. Ultimately, homes are randomly chosen from each SSU [18]. Trained interviewers conduct in-person interviews with women of reproductive age and their households, focussing on demographic factors, reproductive health, child health, and family planning. The data for this study were acquired through the World Health Organization's Health Equity Assessment Toolkit (HEAT) online platform. HEAT is a software application developed by the World Health Organization. It facilitates assessing, analysing, and documenting health disparities data [19]. Our study utilised disaggregated data from the GDHS health indicators available in HEAT. It is crucial to acknowledge that HEAT does not encompass all data from the GDHS; it mainly consists of selected datasets relevant to health equity assessments.

### Outcome measures and dimensions of inequality

The outcome variable in this study was full immunization coverage among children aged two years. A child was considered fully immunized if they had received one dose of BCG (against tuberculosis), three doses of DPT-HepB-Hib (pentavalent vaccine protects against diphtheria, pertussis, tetanus, hepatitis B, and Haemophilus influenzae type b), at least three doses of oral polio vaccine (OPV), and one dose of measles-containing vaccine (MCV1) before their first birthday. Vaccination status was ascertained using vaccination cards, health facility records, and maternal recall during the survey interviews. Six dimensions of inequality were assessed: age, household wealth status, educational attainment, place of residence, child's sex, and subnational region. Following the WHO classification of reproductive age, women aged 15–49 years were included in the study. The sample was divided into two age groups: 15–19 years (adolescents) and 20–49 years (adults), consistent with the WHO reproductive health framework and previous DHS studies [20]. The WHO HEAT categorised women's household wealth status into five classifications based on categorization in the GDHS: poorest, poorer, middle, richer, and richest. This classification is based on wealth quintiles derived from household asset indices. Household wealth is evaluated based on various assets owned by the household, including real estate, livestock, and durable goods. The data is readily accessible in the Demographic and Health Surveys (DHS) and Multiple Indicator Cluster Surveys (MICS) datasets. Four categories of mother's educational attainment were used: no formal education, primary,

secondary, and higher. Place of residence was based on classification of households as urban or rural in line with the definitions provided by the Ghana Statistical Service and the DHS, which categorize areas according to population size and administrative designations determined during the national census [18]. The child's sex was documented as either male or female. Subnational region comprises sixteen administrative regions: Ahafo, Ashanti, Bono, Bono East, Central, Eastern, Greater Accra, North East, Northern, Oti, Savannah, Upper East, Upper West, Volta, Western, and Western North. A summary of Ghana's sixteen administrative regions, including their locations and key socioeconomic characteristics, is provided in S1 Table.

## Data analysis

We utilised the online version of the WHO HEAT, which facilitates the analysis of health inequality data. HEAT provides estimates, confidence intervals (CIs), and summary metrics of inequality, enabling a thorough assessment of disparities and informed conclusions. Estimates and confidence intervals for complete immunization coverage were calculated using the six inequality dimensions within the HEAT software. Four inequality metrics were employed: Difference (D), Ratio (R), Population Attributable Risk (PAR), and Population Attributable Fraction (PAF). Their definitions, formulas, and interpretations are presented in Table 1.

- D (Difference) measures the absolute difference in full immunization coverage between two subgroups.

- R (Ratio) measures the relative difference in coverage between two subgroups.

- PAR (Population Attributable Risk) quantifies the proportion of coverage in the population that can be attributed to a specific risk factor.

- PAF (Population Attributable Fraction) indicates the percentage of total inequality that would be eliminated if the risk factor were removed.

Both D and R are unweighted metrics, focusing solely on the subgroups being compared, while PAR and PAF are weighted metrics, accounting for the population sizes of each subgroup. Absolute metrics (D and PAR) provide explicit measures of inequality magnitude, whereas relative metrics (R and PAF) contextualise disparities within the broader population. The WHO recommends presenting summary metrics in both absolute and relative formats to support policy-relevant interpretation. Detailed formulas and calculations for these metrics are described in previous literature [21–23].

Table 1. Definition, formula, and interpretation of inequality metrics.

| Metric | Definition | Formula/ Calculation | Type | Interpretation |
|---|---|---|---|---|
| Difference (D) | Absolute difference in coverage between two subgroups | $D = Y_{high} - Y_{low}$ | Absolute | Values near 0 indicate equality; positive values favour advantaged groups |
| Ratio (R) | Relative difference in coverage between two subgroups | $R = Y_{high} / Y_{low}$ | Relative | A value of 1 indicates equality; >1 favours advantaged groups; <1 favours disadvantaged groups |
| Population Attributable Risk (PAR) | Absolute reduction in inequality if all subgroups achieved reference level | $PAR = \mu - Y_{ref}$ | Absolute | Indicates potential increase in overall coverage if inequality removed |
| Population Attributable Fraction (PAF) | Relative reduction in inequality if all subgroups achieved reference level | $PAF = (PAR / \mu) \times 100$ | Relative | Expressed as a percentage of total inequality that could be eliminated |

μ denotes the population-level weighted mean coverage across all subgroups.

 

In addition to quantifying inequality magnitude, PAR and PAF provide policy-relevant measures that estimate the potential improvement in full immunization coverage if existing inequalities were eliminated. PAR reflects the absolute number of cases attributable to inequality in the population, while PAF expresses this burden as a proportion of total coverage that could be improved under conditions of equity. In the context of Ghana, these measures are particularly useful for identifying the public health gains that could be achieved through targeted interventions aimed at disadvantaged groups, thereby supporting evidence-based resource allocation and equity-focused immunization strategies.

### Ethics approval and consent to participate

The 2022 GDHS received ethical approval from the Ghana Health Service Ethics Review Committee and the Institutional Review Board of ICF International. Informed consent was obtained from all participants prior to data collection. This study utilised secondary data from the GDHS accessed through the WHO HEAT, which compiles publicly available, anonymised survey data. As the data are de-identified, no additional ethical approval was required.

### Results

Fig 1 illustrates the trends of full immunization coverage among two-year-olds in Ghana from 1993–2022. Overall, coverage increased from 55.5% in 1993 to 71.5% in 2022, although fluctuations were observed over the study periods. Coverage peaked at 77.6% in 2008 before declining to 71.5% in 2022.

Table 2 presents trends in full immunization coverage among two-year-olds in Ghana from 1993 to 2022 across different inequality dimensions. Coverage among children of mothers aged 20–49 years increased from 55.6% in 1993 to 72.2% in 2022, peaking at 78.2% in 2008, while coverage among children of mothers aged 15–19 years was 45.4% in 2022. Across wealth quintiles, coverage improved over time, with coverage among children from the poorest households increasing from 38.2% in 1993 to 67.5% in 2022. Similar upward trends were observed across the remaining wealth quintiles.

Coverage also improved across all educational categories. Among children of mothers with no formal education, coverage increased from 45.0% in 1993 to 65.0% in 2022, while children of mothers with higher education recorded coverage

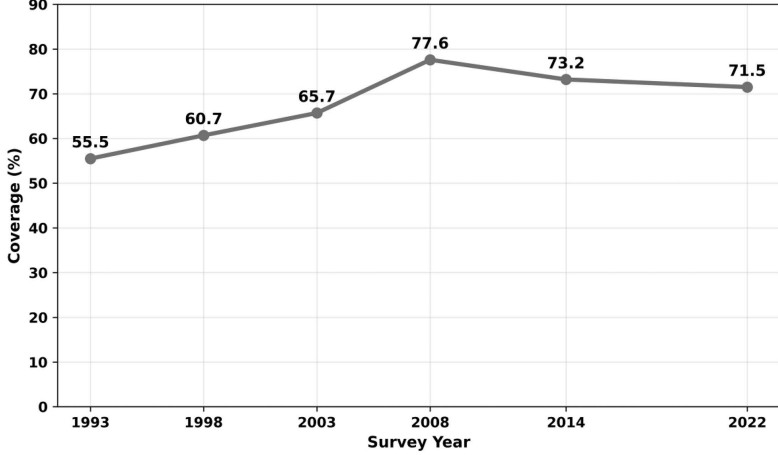

**Fig 1. Trends in the prevalence of full immunization coverage among two-year-olds in Ghana, 1993-2022.**

**Table 2. Trends in the prevalence of full immunization coverage among two-year-olds in Ghana by different inequality dimensions, 1993-2022.**

| Dimension | Measure | 1993 Popu-lation | 1993 Estimate (%) | 1993 CI-LB | 1993 CI-UB | 1998 Popu-lation | 1998 Estimate (%) | 1998 CI-LB | 1998 CI-UB | 2003 Popula-tion | 2003 Estimate (%) |
|---|---|---|---|---|---|---|---|---|---|---|---|
| **Mother's Age (2 groups)** | 15-19 years | NA | NA | NA | NA | NA | NA | NA | NA | NA | NA |
| | 20-49 years | 640 | 55.6 | 51.5 | 59.7 | 536 | 61.1 | 56.5 | 65.5 | 629 | 65.9 |
| **Economic Status (Wealth Quintile)** | Quintile 1 (poorest) | 131 | 38.2 | 29.0 | 48.3 | 131 | 52.0 | 43.1 | 60.8 | 155 | 58.9 |
| | Quintile 2 | 145 | 48.3 | 38.9 | 57.8 | 103 | 54.1 | 43.9 | 64.0 | 132 | 63.3 |
| | Quintile 3 | 141 | 57.4 | 47.6 | 66.8 | 122 | 57.5 | 47.1 | 67.2 | 142 | 64.5 |
| | Quintile 4 | 134 | 65.7 | 57.1 | 73.3 | 106 | 69.5 | 59.3 | 78.0 | 107 | 69.3 |
| | Quintile 5 (richest) | 113 | 70.8 | 63.1 | 77.5 | 87 | 75.8 | 65.7 | 83.7 | 114 | 76.2 |
| **Education (4 groups)** | No education | 260 | 45.0 | 38.5 | 51.7 | 193 | 51.6 | 43.6 | 59.4 | 251 | 59.4 |
| | Primary education | 374 | 61.0 | 56.1 | 65.6 | 117 | 58.7 | 49.0 | 67.7 | 159 | 69.4 |
| | Secondary education | 28 | 78.6 | 58.1 | 90.7 | 237 | 69.0 | 61.8 | 75.3 | 229 | 70.5 |
| | Higher education | NA | NA | NA | NA | NA | NA | NA | NA | NA | NA |
| **Place of Residence** | Rural | 475 | 49.9 | 44.9 | 54.9 | 410 | 58.9 | 53.6 | 64.0 | 425 | 62.4 |
| | Urban | 189 | 69.8 | 63.1 | 75.8 | 139 | 66.3 | 56.6 | 74.7 | 224 | 72.1 |
| **Sex** | Female | 331 | 58.6 | 52.7 | 64.3 | 272 | 58.4 | 52.3 | 64.3 | 336 | 64.7 |
| | Male | 333 | 52.6 | 47.0 | 58.1 | 277 | 63.0 | 56.6 | 69.1 | 313 | 66.9 |
| **Subnational Region** | Ahafo | NA | NA | NA | NA | NA | NA | NA | NA | NA | NA |
| | Ashanti | 123 | 53.7 | 44.1 | 62.9 | 101 | 58.8 | 48.3 | 68.7 | 140 | 66.4 |
| | Bono | NA | NA | NA | NA | NA | NA | NA | NA | NA | NA |
| | Bono east | 69 | 50.7 | 39.3 | 62.1 | 45 | 61.6 | 41.4 | 78.4 | 68 | 73.8 |
| | Central | 67 | 47.8 | 34.6 | 61.2 | 62 | 66.0 | 50.2 | 78.9 | 55 | 59.0 |
| | Eastern | 85 | 58.8 | 45.6 | 70.9 | 93 | 52.3 | 41.4 | 62.9 | 66 | 69.1 |
| | Greater Accra | 53 | 81.1 | 73.7 | 86.9 | 65 | 78.6 | 64.7 | 88.0 | 71 | 61.7 |
| | Northeast | NA | NA | NA | NA | NA | NA | NA | NA | NA | NA |
| | Northern | 71 | 42.3 | 32.4 | 52.7 | 37 | 41.6 | 25.0 | 60.3 | 75 | 44.9 |
| | Oti | NA | NA | NA | NA | NA | NA | NA | NA | NA | NA |
| | Savannah | NA | NA | NA | NA | NA | NA | NA | NA | NA | NA |
| | Upper East | 41 | 63.4 | 43.4 | 79.7 | 16 | 59.5 | 43.0 | 74.1 | 43 | 76.1 |
| | Upper West | NA | NA | NA | NA | 22 | 74.4 | 55.3 | 87.2 | 16 | 83.4 |
| | Volta | 69 | 59.4 | 45.5 | 71.9 | 60 | 55.7 | 42.2 | 68.5 | 42 | 60.6 |
| | Western | 63 | 57.1 | 46.0 | 67.6 | 48 | 64.3 | 49.2 | 77.0 | 74 | 76.9 |
| | Western North | NA | NA | NA | NA | NA | NA | NA | NA | NA | NA |

CI-LB: Confidence Interval Lower Bound; CI-UB: Confidence Interval Upper Bound; NA: Not Available.

of 82.3% in 2014 and 69.2% in 2022. By place of residence, coverage among rural residents increased substantially from 49.9% in 1993 to 73.5% in 2022, while urban coverage remained relatively stable over the later survey years. Coverage improved for both male and female children over the study period.

Regional variations were also observed. Greater Accra consistently recorded relatively high coverage levels across survey years, while the Northern region showed lower coverage throughout the study period, increasing from 42.3% in 1993

| 2003 CI-LB | 2003 CI-UB | 2008 Population | 2008 Estimate (%) | 2008 CI-LB | 2008 CI-UB | 2014 Population | 2014 Estimate (%) | 2014 CI-LB | 2014 CI-UB | 2022 Population | 2022 Estimate (%) | 2022 CI-LB | 2022 CI-UB |
|---|---|---|---|---|---|---|---|---|---|---|---|---|---|
| NA | NA | NA | NA | NA | NA | 45 | 62.6 | 40.0 | 80.9 | 41 | 45.4 | 28.7 | 63.2 |
| 61.7 | 69.8 | 478 | 78.2 | 73.8 | 82.0 | 1046 | 73.7 | 69.7 | 77.3 | 1505 | 72.2 | 69.1 | 75.1 |
| 49.7 | 67.5 | 137 | 70.5 | 60.7 | 78.7 | 217 | 75.6 | 64.2 | 84.3 | 333 | 67.5 | 62.0 | 72.7 |
| 54.6 | 71.2 | 106 | 76.2 | 67.1 | 83.4 | 233 | 75.7 | 68.7 | 81.5 | 337 | 71.1 | 66.0 | 75.7 |
| 55.3 | 72.7 | 91 | 74.3 | 64.4 | 82.3 | 211 | 72.7 | 64.5 | 79.6 | 300 | 73.4 | 66.4 | 79.3 |
| 58.4 | 78.4 | 93 | 85.1 | 75.2 | 91.5 | 209 | 66.0 | 57.3 | 73.7 | 308 | 70.3 | 62.5 | 77.1 |
| 67.5 | 83.1 | 69 | 88.2 | 77.3 | 94.3 | 220 | 75.7 | 66.4 | 83.2 | 267 | 76.4 | 67.6 | 83.3 |
| 52.7 | 65.9 | 154 | 70.7 | 61.3 | 78.6 | 271 | 70.1 | 61.0 | 77.8 | 376 | 65.0 | 59.1 | 70.5 |
| 60.2 | 77.2 | 134 | 79.5 | 71.3 | 85.8 | 239 | 68.6 | 58.7 | 77.1 | 231 | 73.0 | 66.0 | 79.0 |
| 64.0 | 76.2 | 198 | 81.6 | 74.9 | 86.8 | 530 | 76.1 | 71.4 | 80.2 | 807 | 74.5 | 69.9 | 78.7 |
| NA | NA | NA | NA | NA | NA | 50 | 82.3 | 68.5 | 90.8 | 132 | 69.2 | 57.4 | 78.9 |
| 56.9 | 67.5 | 313 | 75.8 | 69.9 | 80.8 | 572 | 75.6 | 68.8 | 81.3 | 760 | 73.5 | 69.8 | 76.9 |
| 65.8 | 77.6 | 183 | 80.9 | 74.2 | 86.2 | 518 | 70.7 | 65.2 | 75.6 | 786 | 69.6 | 64.6 | 74.2 |
| 59.6 | 69.5 | 237 | 75.9 | 69.1 | 81.6 | 524 | 71.8 | 64.6 | 78.0 | 782 | 69.9 | 65.8 | 73.8 |
| 60.4 | 72.7 | 259 | 79.2 | 73.4 | 84.1 | 567 | 74.6 | 69.9 | 78.8 | 764 | 73.2 | 69.0 | 76.9 |
| NA | NA | NA | NA | NA | NA | NA | NA | NA | NA | 36 | 72.0 | 59.5 | 81.8 |
| 58.4 | 73.7 | 93 | 76.8 | 66.6 | 84.7 | 210 | 76.8 | 69.9 | 82.5 | 255 | 72.3 | 63.1 | 80.0 |
| NA | NA | NA | NA | NA | NA | NA | NA | NA | NA | 56 | 79.4 | 68.8 | 87.1 |
| 60.1 | 84.1 | 47 | 84.2 | 68.4 | 92.9 | 92 | 82.6 | 74.4 | 88.5 | 77 | 76.6 | 67.4 | 83.8 |
| 46.5 | 70.4 | 58 | 79.9 | 66.8 | 88.7 | 119 | 59.5 | 42.6 | 74.5 | 152 | 71.4 | 59.8 | 80.7 |
| 56.5 | 79.4 | 41 | 70.3 | 52.8 | 83.4 | 107 | 72.0 | 58.8 | 82.3 | 115 | 71.9 | 57.1 | 83.1 |
| 46.7 | 74.8 | 64 | 93.7 | 83.1 | 97.8 | 178 | 74.6 | 63.5 | 83.2 | 210 | 75.8 | 63.8 | 84.8 |
| NA | NA | NA | NA | NA | NA | NA | NA | NA | NA | 56 | 82.6 | 74.7 | 88.4 |
| 32.7 | 57.7 | 71 | 58.9 | 43.8 | 72.5 | 115 | 64.3 | 45.5 | 79.6 | 168 | 49.7 | 41.9 | 57.5 |
| NA | NA | NA | NA | NA | NA | NA | NA | NA | NA | 48 | 84.2 | 77.9 | 89.0 |
| NA | NA | NA | NA | NA | NA | NA | NA | NA | NA | 52 | 74.2 | 65.1 | 81.7 |
| 62.2 | 86.0 | 23 | 88.5 | 75.5 | 95.1 | 40 | 82.9 | 75.6 | 88.4 | 74 | 57.5 | 45.3 | 68.7 |
| 66.8 | 92.6 | 13 | 77.0 | 58.7 | 88.8 | 26 | 89.6 | 81.7 | 94.3 | 44 | 77.9 | 65.3 | 86.9 |
| 42.9 | 76.0 | 35 | 84.3 | 63.4 | 94.3 | 92 | 80.1 | 69.6 | 87.6 | 60 | 89.5 | 80.3 | 94.7 |
| 64.0 | 86.1 | 52 | 73.0 | 59.3 | 83.4 | 111 | 68.7 | 58.3 | 77.4 | 102 | 68.5 | 53.8 | 80.2 |
| NA | NA | NA | NA | NA | NA | NA | NA | NA | NA | 41 | 78.4 | 69.2 | 85.4 |

to 49.7% in 2022. Interpretation of trends for newly created regions such as Savannah, North East, and Oti is limited due to the absence of data in earlier survey years.

Table 3 presents inequality measures for full immunization coverage among two-year-olds in Ghana from 1993 to 2022. Maternal age-related inequality widened over time, with the difference between children of mothers aged 20–49 years

**Table 3. Inequality measures for full immunization coverage among two-year-olds in Ghana across different inequality dimensions, 1993–2022.**

| Dimension | Measure | 1993 Estimate (%) | 1993 CI-LB | 1993 CI-UB | 1998 Estimate (%) | 1998 CI-LB | 1998 CI-UB | 2003 Estimate (%) | 2003 CI-LB | 2003 CI-UB | 2008 Estimate (%) | 2008 CI-LB | 2008 CI-UB | 2014 Estimate (%) | 2014 CI-LB | 2014 CI-UB | 2022 Estimate (%) | 2022 CI-LB | 2022 CI-UB |
|---|---|---|---|---|---|---|---|---|---|---|---|---|---|---|---|---|---|---|---|
| Mother's Age (5 groups) | D | NA | NA | NA | NA | NA | NA | NA | NA | NA | NA | NA | NA | 11.0 | −10.8 | 32.9 | 26.8 | 8.7 | 45.0 |
| | PAF | NA | NA | NA | NA | NA | NA | NA | NA | NA | NA | NA | NA | 0.6 | 0.6 | 0.6 | 1.0 | 1.0 | 1.0 |
| | PAR | NA | NA | NA | NA | NA | NA | NA | NA | NA | NA | NA | NA | 0.5 | −0.2 | 1.1 | 0.7 | 0.3 | 1.2 |
| | R | NA | NA | NA | NA | NA | NA | NA | NA | NA | NA | NA | NA | 1.2 | 0.8 | 1.7 | 1.6 | 1.1 | 2.4 |
| Economic Status (Wealth Quintile) | D | 32.6 | 20.6 | 44.7 | 23.8 | 11.2 | 36.4 | 17.3 | 5.5 | 29.1 | 17.7 | 5.5 | 29.9 | 0.2 | −13.0 | 13.3 | 8.8 | −0.7 | 18.3 |
| | PAF | 27.4 | 27.3 | 27.5 | 24.8 | 24.6 | 24.9 | 15.9 | 15.8 | 16.0 | 13.6 | 13.5 | 13.7 | 3.4 | 3.4 | 3.5 | 6.8 | 6.7 | 6.8 |
| | PAR | 15.2 | 7.3 | 23.2 | 15.0 | 6.4 | 23.7 | 10.4 | 3.1 | 17.8 | 10.6 | 3.0 | 18.1 | 2.5 | −2.6 | 7.6 | 4.8 | 0.1 | 9.5 |
| | R | 1.9 | 1.4 | 2.4 | 1.5 | 1.2 | 1.8 | 1.3 | 1.1 | 1.6 | 1.3 | 1.1 | 1.5 | 1.0 | 0.8 | 1.2 | 1.1 | 1.0 | 1.3 |
| Education (4 groups) | D | NA | NA | NA | NA | NA | NA | NA | NA | NA | NA | NA | NA | 12.2 | −1.7 | 26.1 | 4.2 | −8.2 | 16.5 |
| | PAF | NA | NA | NA | NA | NA | NA | NA | NA | NA | NA | NA | NA | 12.3 | 12.2 | 12.5 | 0.0 | −0.1 | 0.1 |
| | PAR | NA | NA | NA | NA | NA | NA | NA | NA | NA | NA | NA | NA | 9.0 | −1.4 | 19.5 | 0.0 | −7.5 | 7.5 |
| | R | NA | NA | NA | NA | NA | NA | NA | NA | NA | NA | NA | NA | 1.2 | 1.0 | 1.4 | 1.1 | 0.9 | 1.3 |
| Place of Residence | D | 19.9 | 11.9 | 28.0 | 7.4 | −3.0 | 17.8 | 9.7 | 1.8 | 17.5 | 5.1 | −2.9 | 13.2 | −4.9 | −13.0 | 3.2 | −3.9 | −9.8 | 2.1 |
| | PAF | 25.7 | 25.6 | 25.8 | 9.1 | 9.0 | 9.2 | 9.6 | 9.6 | 9.7 | 4.2 | 4.1 | 4.2 | 0.0 | 0.0 | 0.0 | 0.0 | 0.0 | 0.0 |
| | PAR | 14.3 | 8.4 | 20.2 | 5.5 | −1.4 | 12.4 | 6.3 | 1.4 | 11.3 | 3.2 | −1.5 | 7.9 | 0.0 | −2.8 | 2.8 | 0.0 | −2.2 | 2.2 |
| | R | 1.4 | 1.2 | 1.6 | 1.1 | 1.0 | 1.3 | 1.2 | 1.0 | 1.3 | 1.1 | 1.0 | 1.2 | 0.9 | 0.8 | 1.0 | 0.9 | 0.9 | 1.0 |
| Sex | D | −6.1 | −14.0 | 1.9 | 4.6 | −4.0 | 13.3 | 2.2 | −5.7 | 10.0 | 3.3 | −4.9 | 11.5 | 2.8 | −5.3 | 10.9 | 3.2 | −2.4 | 8.8 |
| | PAF | 0.0 | −0.1 | 0.1 | 3.8 | 3.7 | 3.8 | 1.7 | 1.7 | 1.8 | 2.0 | 2.0 | 2.1 | 1.8 | 1.8 | 1.9 | 2.3 | 2.3 | 2.3 |
| | PAR | 0.0 | −3.8 | 3.8 | 2.3 | −1.8 | 6.4 | 1.1 | −2.7 | 4.9 | 1.6 | −1.9 | 5.1 | 1.3 | −1.2 | 3.9 | 1.6 | −0.6 | 3.9 |
| | R | 0.9 | 0.8 | 1.0 | 1.1 | 0.9 | 1.2 | 1.0 | 0.9 | 1.2 | 1.0 | 0.9 | 1.2 | 1.0 | 0.9 | 1.2 | 1.0 | 1.0 | 1.1 |
| Subnational Region | D | 38.9 | 26.8 | 51.0 | 37.0 | 15.4 | 58.5 | 38.5 | 20.7 | 56.3 | 34.8 | 18.7 | 50.8 | 30.1 | 12.5 | 47.7 | 39.8 | 29.3 | 50.2 |
| | PAF | 44.9 | 44.7 | 45.1 | 29.3 | 29.1 | 29.5 | 26.9 | 26.6 | 27.1 | 20.7 | 20.6 | 20.8 | 22.4 | 22.2 | 22.5 | 25.1 | 25.0 | 25.3 |
| | PAR | 25.1 | 14.4 | 35.9 | 17.8 | 8.0 | 27.6 | 17.7 | −0.7 | 36.0 | 16.0 | 9.4 | 22.7 | 16.4 | 4.7 | 28.1 | 18.0 | 10.1 | 25.9 |
| | R | 1.9 | 1.5 | 2.5 | 1.9 | 1.2 | 3.0 | 1.9 | 1.3 | 2.6 | 1.6 | 1.2 | 2.1 | 1.5 | 1.1 | 2.0 | 1.8 | 1.5 | 2.1 |

CI-LB: Confidence Interval Lower Bound; CI-UB: Confidence Interval Upper Bound; D: Difference; NA: Not Available; PAF: Population Attributable Fraction; PAR: Population Attributable Risk; R: Ratio.

and those aged 15–19 years increasing from 11.0 percentage points in 2014 to 26.8 percentage points in 2022. The ratio increased to 1.6 (95% CI: 1.1, 2.4) in 2022, indicating statistically significant inequality favouring older mothers.

Wealth-related inequalities declined substantially over the study period. The difference between the richest and poorest quintiles decreased from 32.6 percentage points in 1993 to 8.8 percentage points in 2022, indicating reduced economic

disparities; however, these inequalities were no longer statistically significant by 2022. Similarly, educational inequalities narrowed over time and became statistically non-significant in 2022.

Urban–rural inequalities also reduced considerably during the study period, indicating near-equitable coverage by place of residence in 2022. Sex-based inequalities remained minimal throughout the study period and were not statistically significant.

In contrast, regional disparities persisted across survey years. The difference between the best- and worst-performing regions increased slightly from 38.9 percentage points in 1993 to 39.8 percentage points in 2022. The ratio measure remained elevated at 1.8 (95% CI: 1.5, 2.1) in 2022, indicating statistically significant geographic inequalities in full immunization coverage.

## Discussion

This study highlights long-term trends and inequalities in full immunization coverage among two-year-olds in Ghana from 1993 to 2022. From 1993 to 2014, coverage improved from 55.5% to 71.5%, reflecting the improvement of immunization initiatives such as the Expanded Program on Immunization (EPI), which enhanced vaccine accessibility and strengthened healthcare delivery systems [24]. Similar fluctuating trends have been reported in Sierra Leone [25]. However, the subsequent decline in coverage, from 77.6% in 2008 to 71.5% in 2022, highlights persistent challenges beyond the COVID-19 pandemic. These include vaccine supply chain disruptions, such as delayed deliveries, occasional stock-outs, and inadequacies in cold chain management, as well as geographic disparities, particularly in rural and hard-to-reach areas where health infrastructure is limited, and vaccine hesitancy, influenced by cultural beliefs and misinformation [26–28]. Additionally, service delivery constraints such as inadequate in-service training and health system disruption further contributed to stagnation in coverage [27]. The COVID-19 pandemic intensified these challenges by temporarily suspending mass immunization campaigns and reducing attendance at health facilities due to movement restrictions, fear of infection, and resource reallocation to COVID-19 response efforts [29]. According to a recent study from Ghana, as well as estimates from WHO and UNICEF, routine immunization services in Ghana declined in 2020, with coverage for some childhood vaccines dropping by as much as 5–10 percentage points compared to pre-pandemic levels [29–30]. While similar setbacks were observed in other low- and middle-income countries (LMICs), Ghana demonstrated a sharper reduction in immunization coverage inequalities over time compared to countries like Kenya and Côte d'Ivoire [31]. Overall, these findings indicate that while Ghana achieved early gains in immunization coverage, progress has plateaued in recent years, suggesting the need for health system strengthening beyond routine program expansion.

The study results show significant disparities in immunization coverage by maternal age. Children of older mothers aged 20–49 had their coverage improved from 55.6% in 1993 to 72.2% in 2022, peaking at 78.2% in 2008. This aligns with prior research indicating that older mothers tend to have greater knowledge about immunization, better healthcare access, and more autonomy in health decisions [32–33]. However, children of adolescent mothers (aged 15–19) consistently had lower coverage, with only 45.4% fully immunized in 2022. In the Ghanaian context, this disparity reflects a complex interplay of cultural and systemic barriers. Adolescent mothers often face stigma linked to early pregnancy, which can discourage healthcare-seeking behavior due to fear of judgment or discrimination at health facilities [34]. Systemic issues such as limited adolescent-friendly health services, lack of targeted outreach programs, and logistical challenges in rural or underserved areas further restrict access for young mothers [34,35]. Moreover, cultural norms in some Ghanaian communities may limit adolescent mothers' autonomy and decision-making power, reducing their ability to prioritize child health services such as immunization. These barriers contribute to missed immunization opportunities and lower coverage [35]. In contrast, Barchitta et al. found that older mothers were less likely to vaccinate their children [36], suggesting a complex relationship between maternal age and vaccination, highlighting the need for youth-focused maternal and child health interventions.

Immunization coverage for children of mothers in the poorest quintile increased from 1993 to 2022, reflecting the success of efforts to expand immunization access to underserved populations. Similar upward trends were observed across other economic quintiles, suggesting that targeted public health programs and outreach services contribute to reducing socioeconomic barriers [37]. This finding aligns with evidence from Mozambique and Madagascar [38]. However, inequality analysis shows wealth status was not statistically significant, suggesting that observed differences may not be large at the population level. Despite these, economic vulnerability continues to indirectly affect, particularly during crises such as the COVID-19 pandemic, when transportation barriers, communication gaps, and access constraints disproportionately affected poorer households [39,40]. As inequality disfavored younger women in this study, there is a need to improve girl child education and remove financial barriers to accessing healthcare services, including immunizations [41].

Coverage remained relatively higher among children of mothers with higher education compared with those whose mothers had no formal education. These findings suggest an educational gradient in immunization uptake, where mothers with higher education attainment may have greater health literacy and better understanding of immunization schedules and child healthcare practices [42–43]. Conversely, a study in Ghana by Danso and colleagues observed that maternal education level had no significant effect on caregivers' ability to complete the EPI schedule. However, they found that poor maternal knowledge impedes a child's vaccination activities [44]. This may imply that formal education alone does not necessarily translate into understanding vaccination uptake, indicating the need for targeted health education to bridge this gap.

Urban-rural differences showed improved coverage in rural areas, while urban areas remained relatively stable over time. Coverage among children of mothers in rural areas rose substantially from 1993 to 2022, suggesting narrowing urban–rural disparities and relatively improved coverage in rural areas in recent years. Rural improvements are likely driven by expanded outreach services, CHPS implementation, and targeted immunization campaigns [15]. In contrast, urban areas maintained relatively stable coverage rates, suggesting that access to immunization services in urban settings may have plateaued. While urban areas often benefit from better infrastructure, proximity to healthcare facilities, and higher health literacy, persistent issues such as overcrowding, vaccine hesitancy, and socioeconomic inequalities may hinder further improvements [45]. To close this gap, the Ghana Health Service has introduced community-based health planning and services (CHPS) to increase access to primary health care, including immunizations in hard-to-reach areas. The CHPS initiative has played a key role in primary healthcare access, particularly in underserved rural communities [46–47]. However, persistent geographic inequities remain, particularly in Northern Ghana, where infrastructure limitations, distance to facilities, and cultural barriers continue to constrain access.

Sex disparities in immunization coverage were found, although improvements were observed for both sexes over the study period. A slightly higher coverage was observed among males in recent survey years. However, overall differences remain small. This suggests that Ghana's EPI program is broadly equitable by sex, consistent with findings from other LMICs where sex differences are minimal or inconsistent [36,39,48–50]. Nonetheless, continued monitoring is important to ensure no emerging sex bias in service delivery.

Further, regional variations in immunization coverage of two-year-olds were observed across Ghana, with Greater Accra maintaining progress probably due to better healthcare infrastructure, higher health literacy, and easier access to services, often serving as the hubs of immunizations. This finding is consistent with previous studies in Ghana [51] and Gambia [52]. Pythagore et al. reported that high vaccination coverage exists among wealthier families and well-educated individuals, predominantly in urban areas [31]. On the other hand, the Northern part of Ghana exhibited slower progress, which may be attributed to factors such as long distances to vaccination centres, limited healthcare infrastructure, lack of transportation to neighbouring urban health facilities, and cultural barriers to vaccine uptake [51]. Ghana encountered significant challenges in vaccine coverage and measles outbreaks in 2022, with the Savannah Region experiencing a sharp rise in measles cases, largely due to reduced vaccination coverage following the COVID-19 pandemic [53]. These regional disparities emphasize the need for context-specific strategies to address the unique challenges faced in underperforming areas of Ghana.

From a programmatic perspective, equity analyses should also be interpreted alongside population distribution to inform prioritization of interventions. For example, among regions with lower immunization coverage, the Northern Region accounts for approximately 7.6% of the national population compared to 4.2% in the Upper East Region, suggesting that prioritizing interventions in more populous regions may yield greater overall improvements in national immunization outcomes [18]. This approach enables program managers to allocate resources more efficiently and maximize the public health impact of equity-focused immunization strategies.

Furthermore, the PAR and PAF estimates in Table 3 provide additional insight into the population-level burden of these inequalities by quantifying the proportion of immunization coverage gaps that could be eliminated if all regions achieved the level of coverage observed in the most advantaged groups. The relatively higher PAR and PAF values for regional disparities indicate that geographic inequalities contribute substantially to the overall national coverage gap, and that targeted improvements in underperforming regions would yield the greatest gains in national immunization coverage.

### Policy and practice implications

The findings emphasize the need for policy and interventions to strengthen Ghana's immunization programs. Building a resilient health system is necessary to sustain immunization coverage during health emergencies, such as the COVID-19 pandemic. It is essential to develop targeted strategies, including expanded outreach programs, improved healthcare infrastructure, and culturally sensitive approaches, necessary to address regional and socioeconomic disparities. Special attention should be paid to underperforming regions such as the Northern and Savannah areas, where limited access to healthcare, transportation barriers, and cultural factors hinder vaccine uptake. Strengthening community-based initiatives like the CHPS program can enhance immunization access in hard-to-reach areas.

Additionally, improving maternal health education and empowering adolescent mothers is critical to addressing age-related disparities in immunization coverage. Continuous in-service training for healthcare providers can ensure quality service delivery and better vaccine supply chain management. Investments in robust health data systems are also necessary to improve data quality and surveillance, enabling policymakers to monitor progress and address coverage gaps effectively. By implementing these strategies, Ghana can reduce inequalities, sustain immunization gains, and move closer to achieving universal childhood vaccination coverage.

Beyond the Ghanaian context, these findings have broader implications for other low- and middle-income countries facing persistent inequities in immunization coverage. Evidence from similar settings, including Kenya and Côte d'Ivoire, suggests that addressing such gaps requires targeted strategies such as prioritizing zero-dose children, strengthening outreach services in underserved communities, and integrating immunization with primary healthcare systems [13,31,38]. These approaches have been shown to improve equitable vaccine delivery and reduce missed opportunities for immunization. Therefore, the application of equity-focused metrics such as PAR and PAF in this study provides not only context-specific insights but also a framework for guiding resource allocation and intervention prioritization in comparable settings [21–22].

### Strengths and limitations

This study comprehensively analyses full immunization coverage among two-year-olds in Ghana over nearly three decades (1993–2022), offering valuable insights into long-term trends and inequalities. The HEAT systematically evaluates disparities within sociodemographic groups, including maternal education, wealth index, and geographic location. HEAT's ability to provide multiple summary measures of inequality enhances the depth of analysis, ensuring a clear understanding of immunization coverage gaps. Additionally, using data from the DHS strengthens the reliability of the findings, as DHS surveys are nationally representative and provide disaggregated data for meaningful subgroup comparisons. The longitudinal study facilitates trend analysis, identifying progress and persistent challenges.

The study has limitations. The HEAT data analytical processes can be complex and require technical expertise for proper interpretation. HEAT relies on secondary DHS datasets, so it may not capture all factors influencing immunization coverage, such as cultural beliefs, caregiver decision-making processes, or healthcare system-specific challenges. Furthermore, the DHS data are based on self-reported vaccination uptake, introducing the potential for inability to report accurately. Additionally, DHS data are cross-sectional, which limits the ability to infer causal relationships between observed disparities and specific determinants. The underrepresentation of hard-to-reach or marginalized populations in survey samples could also influence findings. Another limitation of this study is that the equity analysis is based on grouped data, where both the explanatory variables (e.g., socioeconomic status and subnational regions) and the outcome variable (immunization coverage) are expressed as group-level averages. As a result, within-group variability is not captured, which may introduce the risk of ecological or group-level correlation fallacy. Therefore, the observed associations should be interpreted at the population or group level and not inferred as individual-level relationships. This limitation should be considered when interpreting the findings, as individual-level heterogeneity within groups may influence immunization coverage patterns. Finally, while the study highlights inequalities, it may not fully explore the contextual reasons driving regional, socioeconomic, or maternal education-based disparities.

## Conclusion

Ghana has made substantial progress in improving full immunization coverage among two-year-olds from 1993 to 2022. However, progress has stalled in recent years, with persistent inequalities particularly related to maternal age and Subnational region. These findings highlight the need for targeted, equity-focused interventions to address remaining gaps and strengthen health system resilience. Future research should explore causal pathways underlying these disparities and evaluate targeted interventions aimed at high-risk populations, including adolescent mothers and underserved regions.

## Supporting information

**S1 Table. Administrative regions of Ghana and their geographical zones and characteristics.**
(DOCX)

## Author contributions

**Conceptualization:** Florence Gyembuzie Wongnaah, Augustus Osborne.

**Data curation:** Florence Gyembuzie Wongnaah, Augustus Osborne.

**Formal analysis:** Augustus Osborne.

**Supervision:** Bright Opoku Ahinkorah.

**Writing – original draft:** Florence Gyembuzie Wongnaah, Khadijat Adeleye.

**Writing – review & editing:** Florence Gyembuzie Wongnaah, Augustus Osborne, Khadijat Adeleye, Bright Opoku Ahinkorah.

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
