## [Decision Letter · Decision Letter 0]

2 Mar 2026

PONE-D-25-55499Progress and gaps in childhood immunization among two-year-olds in Ghana (1993–2022): A trend and equity analysisPLOS One

Dear Dr.  Wongnaah,

Thank you for submitting your manuscript to PLOS ONE. After careful consideration, we feel that it has merit but does not fully meet PLOS ONE’s publication criteria as it currently stands. Therefore, we invite you to submit a revised version of the manuscript that addresses the points raised during the review process.

We look forward to receiving your revised manuscript.

Kind regards,

Bereket Yakob, Ph.D.

Academic Editor

PLOS One

Journal Requirements:

3. Please ensure that you refer to Figure 1 in your text as, if accepted, production will need this reference to link the reader to the figure.

4.We note you have included a table to which you do not refer in the text of your manuscript. Please ensure that you refer to Table 2 and 3 in your text; if accepted, production will need this reference to link the reader to the Table.

Additional Editor Comments:

Dear authors, I agree with the reviewers’ comments and invite you to submit a revised version. I have also added my comments for your review. Please see the specific comments below.

Reviewer's Responses to Questions

**Comments to the Author**

1. Is the manuscript technically sound, and do the data support the conclusions?

Reviewer #1: Yes

Reviewer #2: Yes

2. Has the statistical analysis been performed appropriately and rigorously? 

Reviewer #1: Yes

Reviewer #2: Yes

3. Have the authors made all data underlying the findings in their manuscript fully available?

Reviewer #1: Yes

Reviewer #2: Yes

4. Is the manuscript presented in an intelligible fashion and written in standard English?

Reviewer #1: Yes

Reviewer #2: Yes

5. Review Comments to the Author

Reviewer #1: This is a nicely written article trying to identify some associations between low vaccination coverage and equity. The authors have used the HEAT database that is publicly available. However, they should consider the following points.

1. Although the authors have given the PAR and PAF in Table 3, they have given no description or interpretation of these two measures in either the Results or the Methods sections. They should explain to the reader the utility and practical applications of these measures in the context of Ghana.

2. In Table 1 they should add what “mu” stands for. I have assumed it represents the weighted sample mean.

3. The equity analysis is based on grouped data and the putative determinants or explanatory variables of inequitable coverage (like socio-economic status, subnational area etc.) as well as the response variable are all group average values. A key limitation of such analyses is the group correlation fallacy or ecological fallacy in epidemiology, because the within group variation of explanatory variables are subsumed within the group average values. The authors should mention this in their discussion on limitations.

4. Discussion on some practical use of this research to the programme manager will enhance the usefulness of this manuscript. Authors can reflect the proportion of population affected by a given measure of inequity. For instance, of the two subnational areas with the lowest coverage in 2022 – Northern and Upper East – Northern has ~11% of the population whereas Upper East has only about 5% of the population. Discussing the equity analyses in this context will help the programme manager decide on prioritizing intervention to achieve the maximum impact.

5. In the tables the decimal points for coverage estimates are cluttering up the display. Authors and copy editors may consider rounding off the decimal points.

Reviewer #2: This important and interesting study, conducted under the auspices of various public organizations, including the WHO, assesses the effectiveness of vaccination coverage among children in their first years of life in Ghana. While it is interesting and logically written using appropriate statistical methods, it is essentially presented as a report on the implementation of expanded programs on immunization in a specific region and will be more useful in WHO bulletins and other documents for organizing vaccination programs and identifying gaps. However, it has weak scientific significance. The only significant point I found in the presented manuscript was the fact that, over a long period of observation in this specific region, a gap in support for the group of adolescents, that renew annually, persists. Possible ways to overcome this problem, including examples from other countries, are needed to present this article's scientific significance.

6. PLOS authors have the option to publish the peer review history of their article (what does this mean?). If published, this will include your full peer review and any attached files.

Reviewer #1: No

Reviewer #2: No

---

## [Author Response · Author response to Decision Letter 1]

14 Apr 2026

Response

We thank the reviewer for this important comment. We acknowledge that the previously cited WHO HEAT platform is not an acceptable data repository under PLOS ONE requirements. In response, we have now deposited the minimal analytical dataset required to replicate our findings in a public repository (Figshare), and a DOI has been provided in the revised manuscript. The Data Availability Statement has been updated accordingly.

3. Please ensure that you refer to Figure 1 in your text as, if accepted, production will need this reference to link the reader to the figure.

Response

We thank the reviewer for this helpful suggestion. We have ensured that Figure 1 is referenced in the main text to facilitate proper linking during production.

4.We note you have included a table to which you do not refer in the text of your manuscript. Please ensure that you refer to Table 2 and 3 in your text; if accepted, production will need this reference to link the reader to the Table.

Response:

We thank the reviewer for this helpful suggestion. We have carefully revised the manuscript to ensure that all tables, including Tables 2 and 3, are consistently cited in the main text to comply with journal production requirements.

Response:

We thank the reviewer for this helpful comment. We have revised the manuscript to include a properly formatted caption for Supporting Information (S1 Table) in accordance with PLOS ONE guidelines. We have also ensured that the Supporting Information file is correctly cited in the main text for clarity and consistency.

Response:

We acknowledge the editor’s guidance regarding citation of reviewer-suggested literature and have evaluated all suggested references for relevance. Only those deemed directly relevant to the study have been incorporated.

---

## [Decision Letter · Decision Letter 1]

4 May 2026

Progress and gaps in childhood immunization among two-year-olds in Ghana (1993–2022): A trend and equity analysis

PONE-D-25-55499R1

Dear Dr. Wongnaah,

We’re pleased to inform you that your manuscript has been judged scientifically suitable for publication and will be formally accepted for publication once it meets all outstanding technical requirements.

Kind regards,

Bereket Yakob, Ph.D.

Academic Editor

PLOS One

Additional Editor Comments (optional):

Reviewers' comments:

Reviewer's Responses to Questions

**Comments to the Author**

1. If the authors have adequately addressed your comments raised in a previous round of review and you feel that this manuscript is now acceptable for publication, you may indicate that here to bypass the “Comments to the Author” section, enter your conflict of interest statement in the “Confidential to Editor” section, and submit your "Accept" recommendation.

Reviewer #2: All comments have been addressed

2. Is the manuscript technically sound, and do the data support the conclusions?

Reviewer #2: Yes

3. Has the statistical analysis been performed appropriately and rigorously? 

Reviewer #2: Yes

4. Have the authors made all data underlying the findings in their manuscript fully available?

Reviewer #2: Yes

5. Is the manuscript presented in an intelligible fashion and written in standard English?

Reviewer #2: Yes

6. Review Comments to the Author

Reviewer #2: All previously noted comments have been taken into account by the authors. The corrections are logically presented in the new version of the article. I express my gratitude to the authors for their critical approach to editing these shortcomings.

7. PLOS authors have the option to publish the peer review history of their article (what does this mean?). If published, this will include your full peer review and any attached files.

Reviewer #2: **Yes:** Mikhail P. Kostinov

---

## [Editor Report · Acceptance letter]

PONE-D-25-55499R1

PLOS One

Dear Dr. Wongnaah,

I'm pleased to inform you that your manuscript has been deemed suitable for publication in PLOS One. Congratulations! Your manuscript is now being handed over to our production team.

Kind regards,

on behalf of

Dr. Bereket Yakob

Academic Editor

PLOS One